# Soil moisture modelling with ERA5-Land retrievals, topographic indices and in-situ measurements and its use for predicting ruts

Marian Schönauer[1], Anneli M. Ågren[2], Klaus Katzensteiner[3], Florian Hartsch[1], Paul Arp[4], Simon Drollinger[5], Dirk Jaeger[1]

[1]Department of Forest Work Science and Engineering, University of Göttingen, Göttingen, Germany
[2]Department of Forest Ecology and Management, Swedish University of Agricultural Sciences, Umeå, Sweden
[3]Institute of Forest Ecology, University of Natural Resources and Life Sciences, Vienna, Vienna, Austria
[4]Forestry and Environmental Management, University of New Brunswick, New Brunswick, Canada
[5]Department of Physical Geography, University of Göttingen, Göttingen, Germany

*Correspondence to*: Marian Schönauer (marian.schoenauer@uni-goettingen.de)

## Abstract

Spatiotemporal modelling is an innovative way of predicting soil moisture and has promising applications in supporting sustainable forest operations. One such application is the prediction of rutting, since rutting can cause severe damage to forest soils and ecological functions.

In this work, we used ERA5-Land soil moisture retrievals and several topographic indices to model variations of in-situ soil water content, by means of a random forest model. We then correlated the predicted soil moisture with rut depth from different trials.

Our spatiotemporal modelling approach successfully predicted soil moisture with a Kendall's rank correlation coefficient of 0.62 ($R^2$ of 64%). The final model included the spatial depth-to-water index, topographic wetness index, stream power index, as well as temporal components such as month and season, and ERA5-Land soil moisture retrievals. These retrievals showed to be the most important predictor in the model, indicating a large temporal variation. The prediction of rut depth was also successful, resulting in a Kendall's correlation coefficient of 0.61.

Our results demonstrate that by using data from several sources, including ERA5-Land retrievals, topographic indices and in-situ soil moisture measurements, we can accurately predict soil moisture and use this information to predict rut depth. This has practical applications in reducing the impact of heavy machinery on forest soils and avoiding wet areas during forest operations.

**Keywords:** spatiotemporal modelling, forest management, forest engineering, rutting, downscaling, reanalysis

## 1    Introduction

For decades, forestry research has sought solutions to accurately predict the trafficability of forest soils (Mattila and Tokola, 2019; White et al., 2012; Murphy et al., 2007). In order to further sustainable forest management, efficient protection of forest soils is mandatory (Vega-Nieva et al., 2009; Picchio et al., 2020; Uusitalo et al., 2019). Heavy harvesting and forwarding machines have been frequently associated with severe soil damage, particularly when operating on soils with low bearing capacity (Horn et al., 2007; Allman et al., 2017). Soil compaction is a common consequence of harvesting operations (Ampoorter et al., 2010; Eliasson, 2005; DeArmond et al., 2021) and has been shown to be detrimental to a number of

ecological functions, including soil biota (Beylich et al., 2010), hydrological patterns, and nutrient supply, with potential drawbacks on plant growth and site productivity (Curzon et al., 2022). In addition to soil compaction, machine traffic can also result in deep ruts (Horn et al., 2007; Ala-Ilomäki et al., 2021; Poltorak et al., 2018), which affect site hydrology and increase anaerobic conditions at the rut's base, where air-filled porosity is reduced, leading to minimized soil aeration (Hansson et al., 2019).

The risk of causing high degrees of soil compaction and rutting is mainly attributed to soil properties such as initial soil bulk density and texture, as well as the current soil water content (Cambi et al., 2015; Crawford et al., 2021). Moist soils show a higher susceptibility to damage since the internal friction is decreased through water embracing soil particles (Hillel, 1998), reducing the soil bearing capacity and the ability for elastic responses to machine-induced impacts (McNabb et al., 2001).

To support forestry management and machine operators, accurate cartographic information on soils with low bearing capacity is essential (Jones and Arp, 2017; Sirén et al., 2019; Campbell et al., 2013). However, existing models that rely on detailed soil maps to retrieve soil mechanical parameters (e.g. Grüll, 2011; Heubaum, 2015) require a high level of input data, and high-resolution soil maps are only available for selected areas, hindering their large-scale application (Vega-Nieva et al., 2009; Kristensen et al., 2019). Therefore, researchers have turned to topographic modelling as a more promising approach (Lidberg et al., 2020; White et al., 2012), as it requires only digital elevation models (DEM), which are increasingly available for most parts of Europe (Hoffmann et al., 2022; Guo et al., 2017). One topographic index that has been extensively studied is the "depth-to-water" (DTW) concept, originally developed and tested at the University of New Brunswick by Meng, Ogilvie, and Arp, as described by Murphy et al. (2007; 2009). The DTW concept calculates flow lines across areas of interest by determining a flow accumulation and selecting lines that originate at a set threshold of accumulated upstream contributing areas. Using a cost function that considers the cell-to-cell slopes, the vertical distances from each cell within a raster to the nearest simulated flow line are ascertained. DTW is well documented (White et al., 2012; e.g. Vega-Nieva et al., 2009; Murphy et al., 2011).

Previous research has shown that the DTW index performs relatively well in predicting wet areas in forested formerly glaciated landscapes compared to other indices (Ågren et al., 2014; Larson et al., 2022). Recent studies have explored further developments in moisture prediction by utilizing machine learning algorithms applied to a variety of freely available data and diverse retrieved information, including different topographic indices calculated on DEMs. Ågren et al. (2021) used 28 topographic predictor variables in an eXtreme Gradient Boosting model (Chen et al., 2021) to predict soil moisture across the entire Swedish forest landscape at high resolution (2x2 m). Although topographic modelling approaches are widely used, they often fail to adjust to seasonal changes in soil water regimes. Static maps may not adequately represent temporal occurrences of flow lines, wet fields, or water-saturated soils. To address this issue, the DTW concept offers a potential solution, enabling the calculation of different scenarios ranging from 'very dry' or 'frozen' to 'wet' soil conditions. However, selecting the most accurate DTW scenario requires high expertise (Leach et al., 2017: 5434; Lidberg et al., 2020), and mistakes can lead to reduced accuracy and result in potential soil damages that could be avoided.

Therefore, we believe that the next crucial step in soil moisture modelling is to incorporate a temporal component that enables the prediction of rasters for any given time and area. One approach to achieve this was designed by Schönauer et al. (2022), who developed a spatiotemporal prediction model. Dynamic satellite-based retrievals of soil moisture with coarse spatial resolution (Soil Moisture Active Passive Mission) were combined with high-resolution but static topographic maps. This resulted in improved performance in predicting moisture values across time-series conducted on sites in Finland, Germany,

and Poland. The incorporation of a dynamic component into the prediction model enabled reflection of the current overall
moisture conditions on the study sites. This allowed to calculate daily prediction grids that could support forestry practice and
enable the guidance of machine operators on sites to avoid traffic on wet areas susceptible to damages. However, a validation
of predicting rut depth by models of this kind has not been facilitated yet.
The effectiveness of soil moisture modelling, whether based on static or dynamic independent variables, is ultimately
constrained by the quality of the dependent variable, which in this case is in-situ soil moisture. Manual measurements of soil
moisture have been conducted in numerous studies using different devices, such as hand-held time-domain reflectometry
sensors (Kemppinen et al., 2018; Uusitalo et al., 2019) or impedance measuring techniques (e.g. Schönauer et al., 2021b).
Despite the potential inaccuracies associated with these techniques (Walker et al., 2004; Francesca et al., 2010), they offer
significant advantages in terms of flexibility, scalability, low investment costs, and minimal maintenance. Another option is
the use of continuously measuring sensor networks (e.g. Oliveira et al., 2021), which can provide relatively reliable
measurements but with limited spatial coverage due to the high costs of installation and maintenance.
In this study, we built upon the approach developed by Schönauer et al. (2022) by incorporating additional data sources,
including additional topographic indices, soil maps, and soil moisture retrievals from ERA5-Land for two soil depths. The
study also used two types of data sources for soil moisture measurements: manual measurements using a handheld moisture
meter, and data from two continuously measuring sensor networks. We argue that manual measurements are simpler and can
be applied to larger areas, while sensor networks are more expensive and limited to chosen positions.
The study had two main objectives: 1. to train soil moisture models using the two individual data sets (manual measurements
and sensor networks) and evaluate their prediction performance, and 2. to select the best combination of predictor variables
(e.g. topographic indices, ERA5-Land values) using a repeated cross-validation approach and compare the best models with
rut depth data obtained during four trials using a forwarder.

## 2    Material and Methods

To model soil water content (SWC), random forest models were trained using two separate datasets: manual in-situ
measurements using an impedance measuring technique (IMT) and continuously measuring soil sensor networks (SSN). To
both datasets we added predictor variables derived from topographic indices (e.g. depth-to-water, topographic wetness index),
soil maps, SWC estimates from the ERA5-Land campaign ($SWC_{ERA}$), and numerical values for date (month and season). We
performed cross-validation and reduced features stepwise to choose the best-performing model. Subsequently, the two final
models (for IMT and SSN) were used to predict SWC for the positions and dates of different field trials with a forwarder.
During these field trials, rut depth data was captured, and compared to the predictions from the final SWC-models (Figure 1).

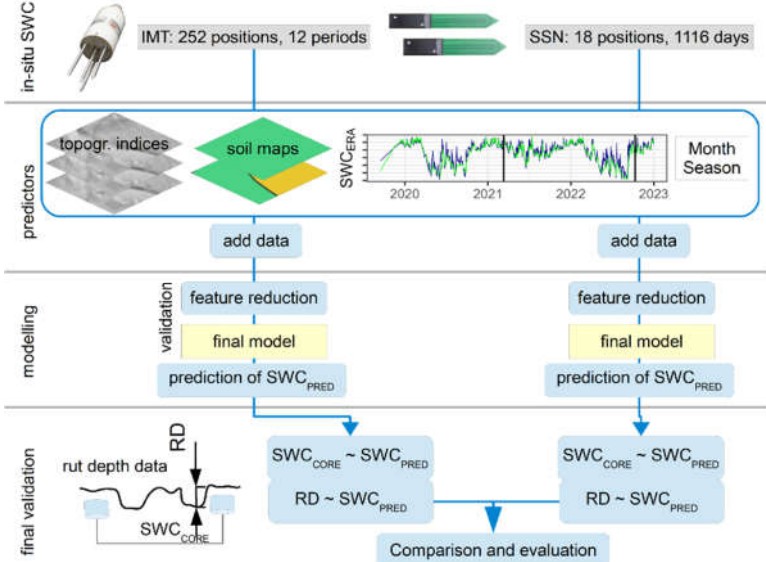

Figure 1: Soil water content (SWC, [%]) was predicted using models trained on two datasets: in-situ measurements (IMT) and soil sensor networks (SSN). Input variables included topographic indices, soil type data, SWC estimates from ERA5-Land ($SWC_{ERA}$), and date values. Through cross-validation, we selected the final models, used to predict $SWC_{PRED}$ for various positions and dates during trials with a forwarder. Model estimates were compared with in-situ $SWC_{CORE}$ and rut depth (RD, [cm]).

**2.1    Study sites**
The data acquisition of volumetric SWC [%] and the trials with a forwarder were conducted in two forest stands located near
the city of Arnsberg in North Rhine-Westphalia (Figure 2). The forest stands were situated at an altitude of approximately
250 m on common soil types such as Cambisol and Stagnosol on Claystone and Sandstone from Devon and Carbon (Table

107    1).

Table 1. Characteristics of the study sites, where soil water content was captured and field trials with a forwarder were performed.

| Site | Coordinates in WGS84 | | Dominant soil types | Humus form | Slope [%] | Canopy |
|------|-----|-----|------|------|------|------|
| | x | y | | | | |
| A | 8.039 | 51.406 | Cambisol - Stagnosol | Mesomull | 15-30 | *Fagus sylvatica, Quercus spp., Pinus sylvestris* |
| B | 8.024 | 51.473 | Stagnosol | Mull | 1-7 | *Fagus sylvatica* |

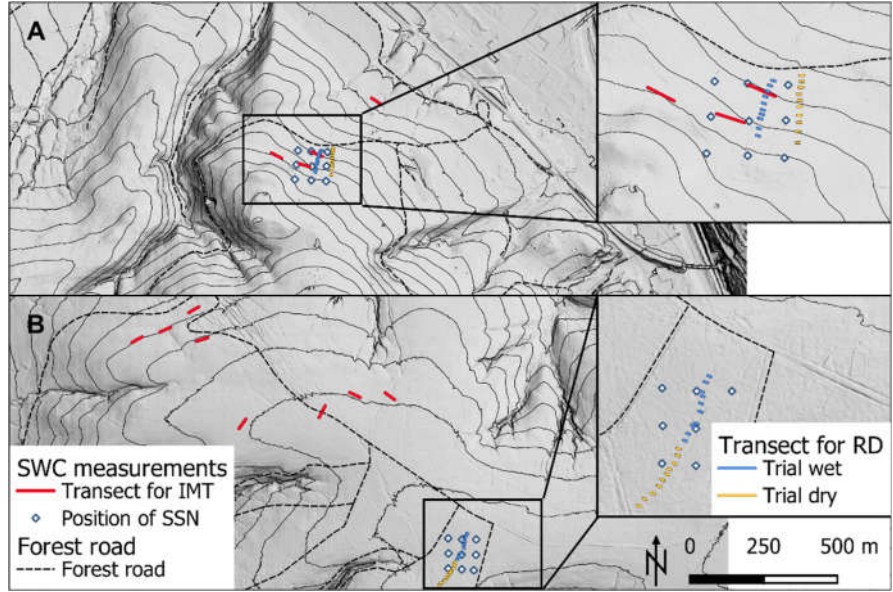

Figure 2: The map indicates the locations of two experimental areas on a hill-shaded digital elevation model with 10 m contour lines; Site A (A, coordinates x, y in WGS84: 8.039, 51.406) and Site B (B, coordinates: 8.024, 51.473), which were used for collecting time-series data on soil water content (SWC). SWC was measured using a handheld soil moisture meter (impedance measuring technique, IMT) along transects (red lines), each containing 21 measuring positions (2 m spacing). In addition, a soil sensor network (SSN) was used to continuously capture SWC at 18 positions (white rhombus). The map also indicates the locations of 40 transects (in crop-outs) used for measuring rut depth (RD) during relatively wet conditions (Trial$_{WET}$, blue lines) and drier conditions (Trial$_{DRY}$, orange lines).
**2.2    Soil moisture models**
**2.2.1    In-situ soil moisture**
Two sets of in-situ data of soil moisture were used: 1. Manual measurements of SWC were performed using a HH2 Moisture
Meter (Delta-T Devices Ltd, England), which applies Impedance Measuring Technique (i.e. 'IMT') (Eijkelkamp Agrisearch
Equipment, 2013). 2. Data from a continuously measuring Soil Sensor Network (i.e. 'SSN').
The IMT data used for this study were previously used for the validation by Schönauer et al. (2022) and consisted of 12
measuring transects. The transects were placed in various positions in broadleaved forests, known to be temporarily wet or
sensitive for machine traffic, with each transect having a length of 40 m. SWC was measured with a spacing of 2 m along the
transects. To measure SWC, measuring rods of 60 mm length were vertically inserted into the soil after removing the humus
layer. The measurements were taken almost monthly between September 2019 and October 2020 (Figure 3B). The IMT data
consisted of 2,184 observations. Overall, this dataset offers a relatively high level of spatial granularity, with 252 measuring
positions. However, the temporal resolution of the data is relatively low, with only monthly measuring campaigns conducted.
The SSN was launched in December 2019 and its data was obtained from continuously measuring SMT100 sensors
(TRUEBNER GmbH, Germany), placed on two sites, each having 9 positions with a spacing of 50x50 m. At each position,
two sensors were placed at a depth of 10 cm in the mineral soil, with a temporal resolution of 15 minutes. The data from these
sensors were averaged for each position and each of the 1,116 days captured (data until 2022-12-31 was included), resulting
in a total of 16,351 observations after omitting all missing values. While this data set provides a high level of temporal
granularity, it suffers from a low level of spatial granularity due to the limited number of positions sampled.
To enable the incorporation of seasonal effects in the modelling approaches, we transformed the date of each measurement
into numeric vectors, resulting in the variables Month and Season. The coding used for Season was as follows: 1 for March,
April, and May; 2 for June, July, and August; 3 for September, October, and November; and 4 for December, January, and
February.
To enable the creation of spatiotemporal data, the positions of all measuring locations were captured using post-processed
signals from a GNSS device (Trimble R2 RTK Rover, Trimble, Colorado, USA). This data was then fused with a range of
topographic indices. To achieve this, values of several topographic indices were extracted at each measuring position of IMT
and SSN.

### 2.2.2 Topographic indices

For calculating topographic indices, we used a freely available digital elevation model (DEM), as provided by the
Bezirksregierung Köln (2020). The resolution of this model was 1x1 m, with a vertical accuracy of ± 0.2 m. Using the free
programming language R (version 4.0.2, R Core Team, 2023) and RStudio (version 2022.07.2, Posit PBC, Massachusetts,
USA), along with the package "rgrass" (Bivand, 2021) to utilize GRASS GIS (Awaida and Westervelt, 2020) commands in
the R interface, the command 'r.hydrodem' was used to 'remove all sinks' (Flags: -a) from the DEM. Thereafter, we calculated
depth-to-water (DTW) maps. To generate these maps, we followed the script by Schönauer and Maack (2021) and used flow
initiation areas (FIA) of the following sizes 0.25 ha (DTW025), 1.00 ha (DTW1), and 4.00 ha (DTW4), which account for
different overall soil moisture conditions. A smaller FIA results in a DTW map for wetter conditions, as the network of
simulated flow lines expands, while a larger FIA represents drier conditions. For further details, refer to Murphy et al. (2009;
145  2011).

The Topographic Wetness Index (TWI) represents the tendency for water to accumulate at any point in the catchment (Quinn
et al., 1991), while the stream power index (SPI) represents the power of water flow at any point in the catchment and the
gravitational forces that move water downslope (Moore et al., 1991). To compute TWI, we used the 'r.watershed' command
in GRASS GIS, as conceived by Sørensen and Seibert (2007). TWI was calculated as $\ln(\alpha/\tan(\beta))$, where $\alpha$ is the cumulative
upslope area draining through a point per unit contour length, and $\tan(\beta)$ is the local slope angle. SPI was calculated as $\alpha *$
$\tan(\beta)$ (Moore et al., 1991). Flow Accumulation, representing the absolute amount of overland flow passing through each cell
was also included as a variable. TWI, SPI, and Flow Accumulation were calculated on an aggregated DEM with a spatial
resolution of 15x15 m. This resolution has been shown to exhibit a stronger correlation with SWC, and can be assumed to be
more robust (Ågren et al., 2014), as observed in prior work where resolutions ranging from 1 to 20 m were tested (data not
shown). In addition, we calculated the variable Slope [°] using the R-package 'raster' (Hijmans, 2020).

### 2.2.3 Soil maps

Soil maps of North Rhine-Westphalia were originally generated at a scale of 1:5,000 from forest site surveys. We included
soil type information (Soil05) for the analysis. While these maps are not available across the entire region of North Rhine-
Westphalia, they were provided for the study sites by the Geological Survey of North Rhine-Westphalia. By contrast, soil
maps with a scale of 1:50,000 are available for the entirety of North Rhine-Westphalia (Soil50).

### 2.2.4 Temporal soil water content from ERA5-Land

ERA5-Land is a global reanalysis dataset providing hourly estimates of meteorological variables at a spatial resolution of 9x9
km, including soil moisture [$m^3$ $m^{-3}$] at the top soil layer (0-7 cm, 'layer 1' (L1)) and at a depth of 7-28 cm ('layer 2' (L2)).
ERA5-Land data is retrieved by assimilating satellite and atmospheric forcing (Muñoz-Sabater et al., 2021).
We opted for ERA5-Land retrievals to address the temporal component of SWC, as this dataset offers a dependable
representation of soil moisture values and their variations across global regions, rendering it suitable for various geophysical
applications (Lal et al., 2022). Additionally, this decision is grounded on two key assumptions: 1. The spatial variability of
SWC is relatively low compared to its temporal variability. 2. The spatial extent of our measurement locations is small and
cannot be adequately captured by satellite-based Earth observation data. Even Sentinel-1, a mission within the Copernicus
Programme by the European Space Agency renowned for supporting high-resolution (1x1 km) surface soil moisture product
generation (Peng et al., 2021), would have limited utility in providing spatial information for our study sites. For instance, the
maximum distance between rut depth transects (Section 2.3.2) was 200 m. Furthermore, since Sentinel-1 focuses on surface
soil moisture using the C-Band, we assume that ERA5-Land's soil moisture estimates for deeper layers might offer a better
fit for our data, as suggested by similar findings presented by Fjeld et al. (2024).
We utilized the API provided by CDS (Copernicus Climate Change Service, 2019) and the R-package 'ecmwfr' (Koen Hufkens
et al., 2019) to download daily grids (at 14:00 UTC) of layer 1 and 2. The downloaded data covered both the whole time span
of our data and the two measuring sites. Both sites were situated in one 9x9 km raster cell of the ERA5-Land. The land cover
for this cell was derived from Bezirksregierung Köln (2023), showing that open land (e.g. grassland, crops) dominated with
52% of the total cover, whereas forests occurred on approximately 31% of the cell size, followed by 12% coverage from
infrastructure, 3% loose material, and 2% water bodies.
After downloading the data, we stacked the daily grids and extracted the corresponding values at each measuring position,
giving $SWC_{ERA}L1$ and $SWC_{ERA}L2$.
All data, the topographic information, soil types, numerical values of date and the dynamic variables from ERA5-Land were
merged with in-situ data, either IMT or SSN.
**2.2.5    Modelling**
The modelling approach described here was applied separately for both data sets, IMT and SSN (and for both datasets
combined).
Initially, we fitted a linear model with SWC as the dependent variable and $SWC_{ERA}L1$, $SWC_{ERA}L2$, Month, Season, DTW025,
DTW1, DTW2, DTW4, Slope, TWI, SPI, Accumulation, Soil05, and Soil50 as the independent variables. We then used this
linear model to check the data for autocorrelations and subsequently eliminated variables with a variance inflation factor > 10
through an iterative process, reducing one variable at a time. Also, the feature selection according to the Boruta algorithm
(package 'Boruta', Kursa and Rudnicki, 2010) was applied.
We then trained random forest models (Breiman, 2001), repeatedly reported as efficient in predicting complex data (Cavalli
et al., 2023; Carranza et al., 2021; Kemppinen et al., 2018), using the 'ranger' package (Wright and Ziegler, 2017) with a 10-
fold cross-validation with 5 repetitions. For each of the 50 models in the validation of one configuration, we noted the mean
of Kendall's coefficient of correlation τ (since different sample sizes occurred) of the random forests and the representative
standard deviation. In addition, the least important variable according to impurity and its frequency within the 50 validation
sets were traced. The variable noted most frequently as least important was then removed, and a new cross-validation was
performed on SWC ~ (n-1) variables, with n being the number of predictors in the model trained previously. This process was
repeated until only one predictor variable remained.
To avoid temporal autocorrelations at the measuring positions, positions IDs were used to select the folds of the cross
validations.

### 2.2.6    Selection of the final model

To select the final random forest model for each data partition, we examined the maximum $\tau$ values obtained and multiplied
them by 0.99 (according to Hauglin et al. (2021)). This was done to penalize the use of an unnecessarily high number of
predictor variables. We selected the model with the least number of predictor variables within this 1%-range as the final
model. The final models (built on IMT and SSN data) were then used to predict rasters of $SWC_{PRED}$, which were visually
evaluated. Subsequently, the outputs of the final models were compared to rut depths and SWC at the machine operating
trails.

## 2.3    Data from field trials with a forwarder

### 2.3.1    Rut depth (RD)

During the field trials conducted in two forest stands at two seasons, a fully loaded forwarder (John Deere 1210G, 8-Wheel
model, total mass of 28 Mg (18 Mg machine weight + 10 Mg loading)) was used. The first trial was conducted on section 1
of an existing machine operating trail on 2021-03-11, during generally wet conditions ($Trial_{WET}$). The second trial was
conducted on subsequent section 2 of the same machine trail on 2022-10-11, during drier conditions ($Trial_{DRY}$) (Figure 2, Site
B), or in close proximity of section 1 (Site A), as there the machine trail was not long enough for both sections.
The four trials were positioned near the sensors of the SSN (Figure 2) and, in the case of Site A, near the IMT measuring
transects. On Site B, the IMT transects were at a distance of 530 m to 1300 m. Moreover, there is a temporal lag between the
IMT measuring campaigns and the field trials (Figure 3). This discrepancy stems from the IMT data being collected as part
of a separate research project.
The 8-wheel machine trafficked section 1 and 2 of both operating trails, and made four passes. Before the first machine pass,
the initial surface was captured along 10 perpendicular transects on each of the four sections. These 4 m wide transects were
placed and marked permanently with inserted wooden pegs. The same pegs were used to position the beam, which served as
the reference height to measure profiles along each transect. Into this beam, metric scales were inserted with a spacing of 10
cm in between, to note the distance between the surface and the beam to the nearest cm. These measured distances (D0, [cm])
describe the surface along the transect on already existing machine operating trails, prior to the trial conducted in this study.
The same procedure was repeated after the fourth consecutive machine passes, giving D4 [cm].
Next, the differences between D0 and D4 were calculated at each of the 41 measurements (10 cm spacing over 4 m) along a
transect. The maximum value of these differences, measured at the left or right machine track, was used to determine rut depth
(RD, [cm]). We used average values of both tracks to prevent pseudo replicates, since intraclass correlation coefficient was
high (0.83), when left and right tracks were integrated separately. Moreover, mean and maximum values of rut depth were
highly correlated (adj. $R^2 = 0.96$).
Four of the 40 transects for measuring RD were not ascertainable as the forwarder destroyed the wooden pegs that positioned
the reference beam. In $Trial_{WET}$, conducted in March 2021, $SWC_{ERA}L1$ and $SWC_{ERA}L2$ showed a soil moisture level of 39%.
At Site A, the measured RD was 10.3±1.9 cm, while at Site B, the RD was 12.7±5.5 cm, with the highest value of RD recorded
after 4 passes, with a depth of 21.5 cm. In Trial$_{DRY}$, conducted in October 2022, the soil water content from ERA5-Land was
32%. At Site A, the measured RD was 3.5±1.7 cm, and at Site B, the RD was 4.3±1.2 cm. Comparisons of RD with DTW
and TWI are given in Figure C1.

**2.3.2    Soil water content at the rut depth transects (SWC$_{CORE}$)**

Volumetric soil moisture content was captured outside the 1$^{st}$, 4$^{th}$, 7$^{th}$ and 10$^{th}$ transect of each section, with a distance of 1 m
to the left and right track, at a depth of 10-15 cm. This water content was determined using 100 cm³ cores taken with an
undisturbed core sampler, with three replicates at each measurement. SWC$_{CORE}$ was calculated according to equation (1):

$$SWC_{CORE}[\%] = \frac{M2 - M1}{M1} * 100 \tag{1},$$

with M2 being the fresh mass of the soil taken with undisturbed cores and M1 being the mass after drying the samples in oven
with 105 °C, until mass constancy was reached.
Measurements of RD and SWC$_{CORE}$ were georeferenced using the GNSS devise and complemented with all the predictor
variables, as described above.

**2.4    Comparisons between model predictions and RD or SWC$_{CORE}$**

For the 'testing on rut depth data' (Figure 1), values of SWC$_{PRED}$ were compared to RD or soil water content, captured through
undisturbed cores along the transects, SWC$_{CORE}$. The predictor variables from the rut depth dataset were used to predict
SWC$_{PRED}$ by means of the final random forest models created in the soil moisture modelling. Since the goodness-of-fit
between in-situ values of RD or SWC$_{CORE}$ and SWC$_{PRED}$ was to some degree sensitive to the seed set during modelling, we
repeated the predictions ten times and used average values to receive robust estimates of SWC$_{PRED}$. To test the correlations
between paired samples of SWC$_{CORE}$ or RD and SWC$_{PRED}$, Kendall's rank correlation was used. We illustrated the
corresponding p-values as follows: `***` for $p<0.001$, `**` for 0.001-0.01, `*` for 0.01-0.05, (`*`) for 0.05-0.10 and 'ns' for p-
values being higher than 0.10. Root mean squared error (RMSE) and mean squared error (MSE) ware calculated according to
Hamner and Frasco (2018). Values are given as mean±standard deviation.

**3    Results**

**3.1    Soil water content**

The mean value of SWC, measured using a handheld moisture meter (IMT), varied between 13.0±10.0% in August 2020 and
43.2±5.95% in February 2020 (Figure 3). Daily mean values obtained from soil sensor networks (SSN) were similar to those
obtained from IMT, ranging from 13.8±2.90% in September 2020 to 39.1±6.66% in March 2020, in the period that
corresponds to the one covered by IMT. The driest conditions were observed in September 2022, with a daily mean SWC of
12.7±2.55%. Overall, the results suggest that IMT and SSN provide comparable estimates of SWC, with the latter providing
higher temporal resolution at a low spatial granularity.

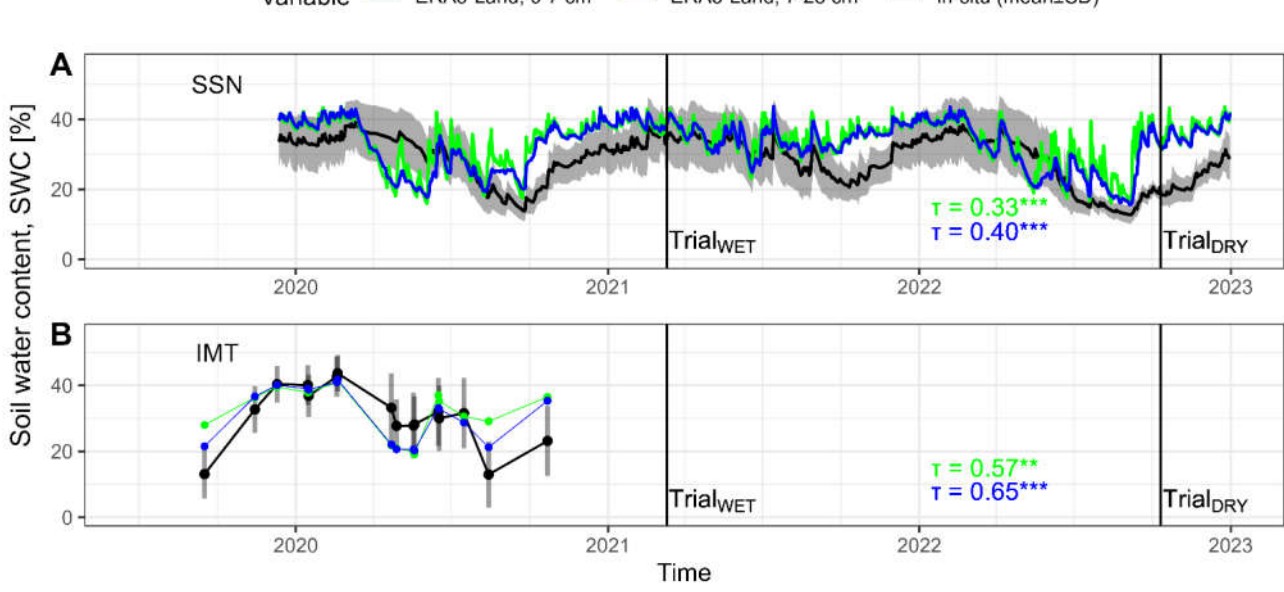

Figure 3: Time series of soil water content (SWC) measured using a soil sensor network SSN (A) with 18 measuring positions on two sites and manual measurements, using impedance measuring technique IMT (B) conducted on 252 positions (black lines/points show daily mean values, grey shading/bars show standard deviation for each day). SWC retrievals from ERA5-Land are shown as a blue line/point (0-7 cm vertical resolution, as available from Copernicus Climate Change Service (2019)) and a green line/point (7-28 cm vertical resolution). The goodness-of-fit between daily means of measured SWC and ERA5-Land retrievals is reported using Kendall's rank correlation coefficient ($\tau$). Vertical lines indicate the dates of the trials when a forwarder conducted four passes at existing machine operating trials.

## 3.2    Soil moisture models

The positions IDs were used to select the 10 folds for cross-validation. However, the dataset SSN had only 18 measuring positions (where SWC was measured on 1116 days), resulting in relatively high deviations of Kendall's $\tau$ of the random forests. The most important feature for this dataset was given by DTW025, although the resulting quality was low, with $\tau$ of 0.363±0.198. By adding the temporal component Month, the $\tau$ improved to 0.637±0.065, which had the lowest standard deviation for the repeated folds. The final model for this dataset included the temporal variables Month and $SWC_{ERA}L2$, as well as the topographic predictor variables TWI and DTW025 (Figure 4). The resulting $\tau$ was 0.710±0.095, revealed through the cross-validation.

For the IMT partition, which had a low temporal but high spatial resolution, the most important feature was the temporal information $SWC_{ERA}L2$, leading to a $\tau$ of 0.569±0.036. The final model had an $\tau$ of 0.620±0.016, including the predictor variables $SWC_{ERA}L2$, Month, Season, and DTW025, TWI, SPI and DTW4.

The main outputs when both datasets were combined can be seen in Figure A1.

279

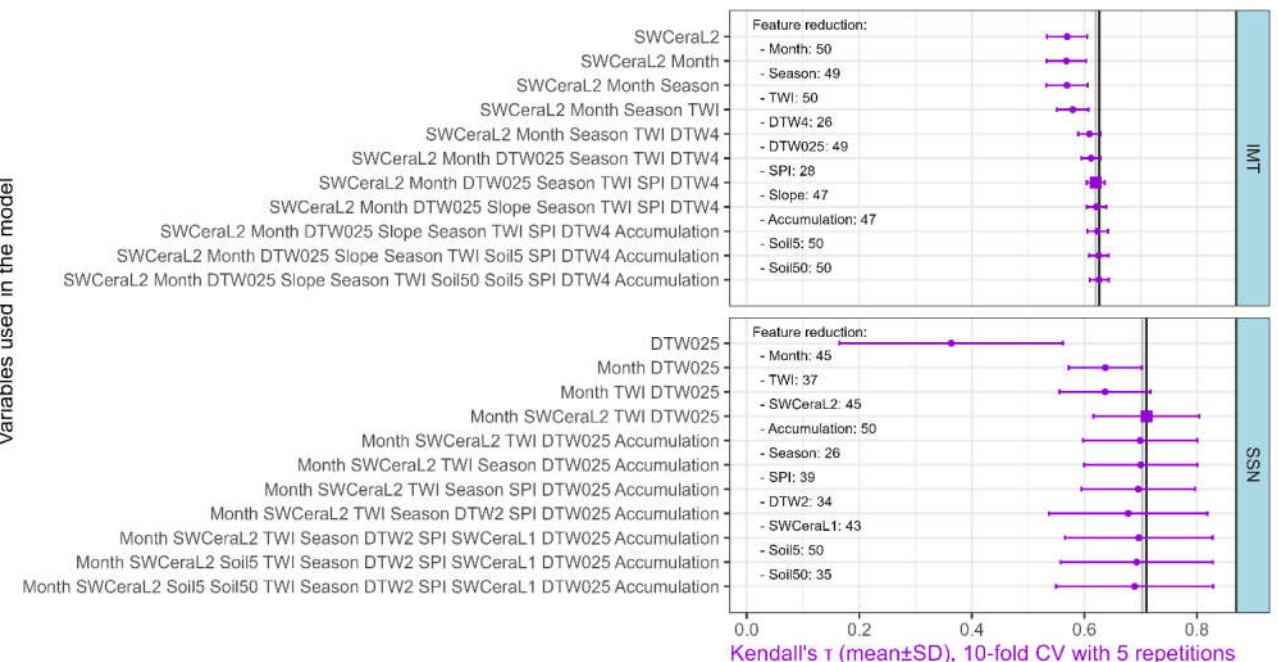

Figure 4: Soil water content (SWC) was modelled by random forests (RF), and evaluated by a repeated 10-fold cross validation (CV). Mean values and standard deviation of resulting values of the Kendall rank correlation coefficient τ during the CV are shown. A stepwise elimination of the least important variable was performed, and the frequency of this variable over all models is provided ("Feature reduction"). The vertical lines indicate the maximum value of τ (black) and the 99% of the maximum (grey), to select final models (squares). Variables used are described in section 2.

### 3.2.1 Comparisons of SWC$_{CORE}$ with SWC$_{PRED}$

The final random forest models of both, the IMT and SSN dataset, were used to calculate SWC$_{PRED}$ on the predictor variables of the rut depth data, including SWC$_{CORE}$ measured at the outside of a subsample of the measuring tracks by undisturbed cores. The comparison between SWC$_{CORE}$ and SWC$_{PRED}$ values predicted by the final random forest models of both datasets (SSN and IMT), revealed a significant association (Figure 5).

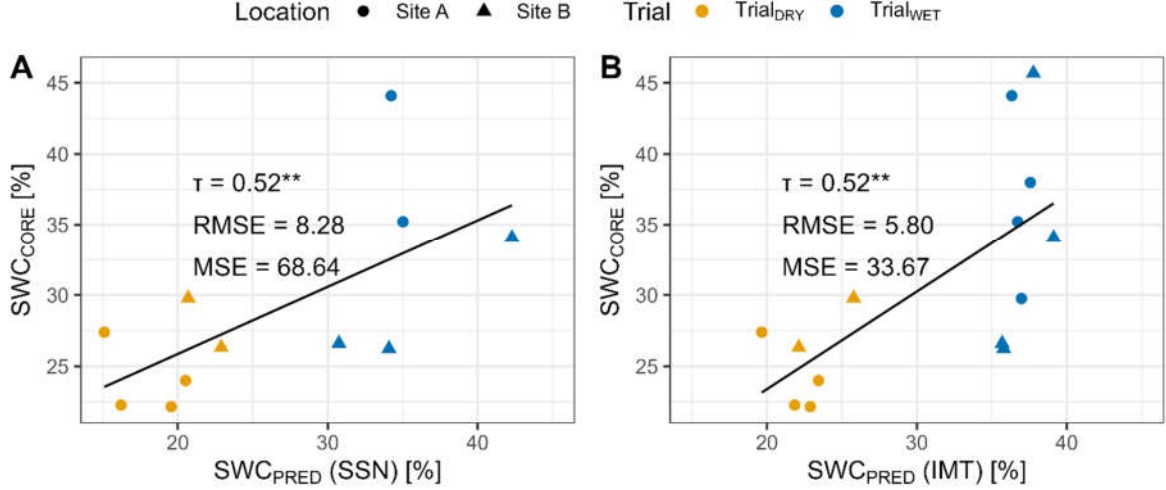

Figure 5: Soil water content was measured during two trials with a forwarder along a machine operating trail (n=14), using 100 cm³ undisturbed cores (SWC$_{CORE}$), and compared to values predicted (SWC$_{PRED}$) by a model trained data from a continuously measuring soil sensor network (SSN, A), or manual measurements with a handheld moisture meter (IMT, B). Correlations were evaluated

using Kendall's τ and significance levels are indicated by \*\*\* for p<0.001, \*\* for 0.001-0.01, \* for 0.01-0.05, (\*) for 0.05-0.10, and 'ns' for p>0.10.

### 3.3     Comparisons of RD with SWC$_{CORE}$ and SWC$_{PRED}$

RD was positively correlated with SWC$_{CORE}$ when both trials with different moisture conditions were included in testing (Figure 6A). However, when each trial was tested separately, no correlation between RD and SWC$_{CORE}$ was observed (Figure B1). Compared to the correlation between RD and SWC$_{CORE}$, modelling outputs SWC$_{PRED}$ proved to be a better predictor of rut depth, particularly for Trial$_{WET}$. The final models that were selected for both datasets produced a Kendall's τ of 0.61 (for IMT, Figure 6B, and SSN, Figure 6C), when comparing RD of the four trials with the corresponding SWC$_{PRED}$. Although the R² values for these models were in similar range (0.620 fot IMT and 0.549 for SSN), we chose to use Kendall's τ since different sample sizes were involved in the analysis. This was particularly relevant for comparing RD with SWC$_{PRED}$ for each Trial separately. While no correlation could be found for Trial$_{DRY}$, correlations were found for Trial$_{WET}$, with Kendall's τ of 0.344 (p=0.037) and 0.281 (p=0.090), for the final models trained on IMT and SSN, respectively (Figure 6B,C). Yet, these correlations seem fragile, as a difference of a few percent of predicted SWC$_{PRED}$ (IMT) is associated with the range of RD between 6.5 and 21.5 cm. Moreover, when analysing the sites separately, a vage trend between SWC$_{PRED}$ and RD could be observed, but without showing significant correlations (Figure B1).

Since the final model trained on IMT data performed slightly better in Trial$_{WET}$ compared to the model trained on SSN data (Figure 6), we chose the IMT model for the generation of prediction rasters for the days of interest (Figure 6B1,B2).

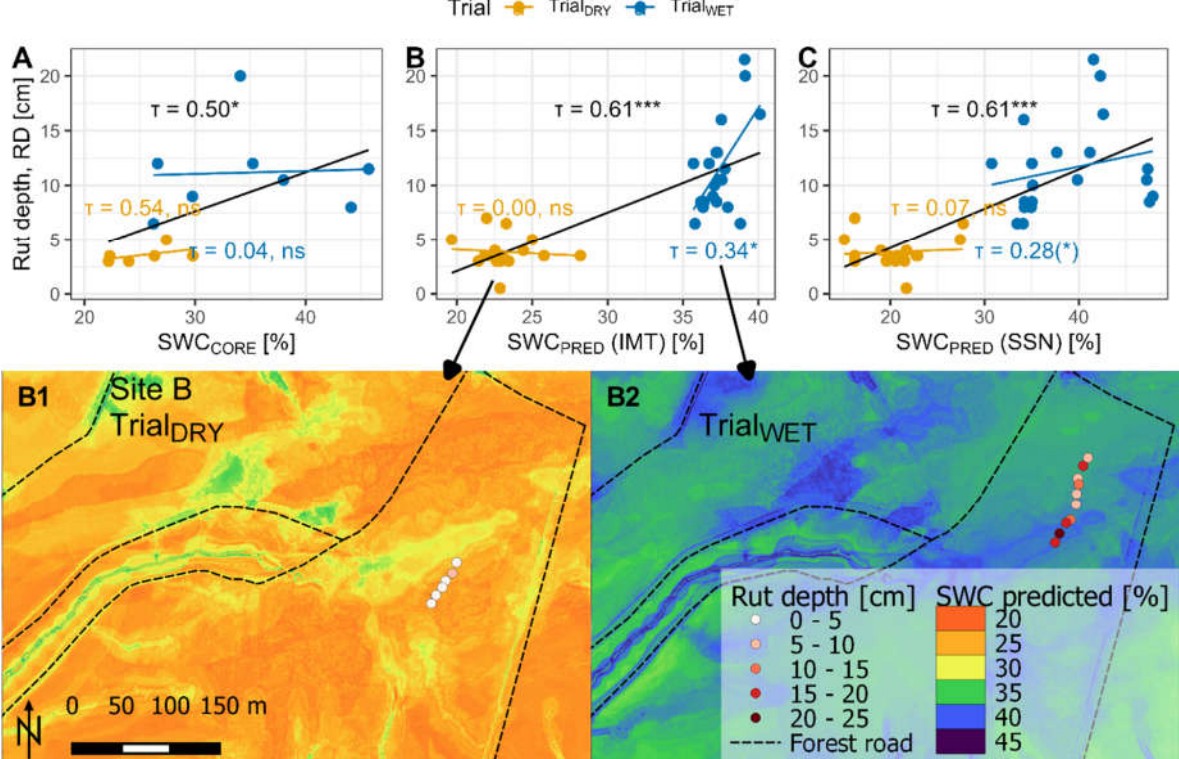

Figure 6: Rut depth (RD) was determined after four passes of a forwarder, driving on two Sites, during two conditions ($_{WET}$ and $_{DRY}$). RD was compared to SWC values, determined for undisturbed soil cores (A) and SWC values predicted by a random forest model trained on manually obtained IMT measurements (B, see Figure 1) and predicted by a model trained data from a continuously

measuring soil sensor network (SSN, C). Correlations were evaluated using Kendall's τ. The correlation of all values is given in black, blue and yellow show the Trials during wet and dry conditions. Significance levels are indicated by ∗ ∗ ∗ for p<0.001, ∗ ∗ for 0.001-0.01, ∗ for 0.01-0.05, (∗) for 0.05-0.10, and 'ns' for p>0.10. The model based on IMT data (B) was used to calculate prediction rasters for the days of the field trials (B1, B2).

## 4    Discussion

### 4.1    Importance of predictive systems

Wet soils are prone to soil disturbances like the formation of deep ruts (Poltorak et al., 2018; McNabb et al., 2001), since water implies a reduction of particle-to-particle bondings within the soil (Hillel, 1998), decreasing the resistance to external forces. Consequently, accurate predictions of soil water content (SWC) and soil trafficability is essential for sustainable forest management and cost-effective, environmentally friendly harvesting operations (Vega-Nieva et al., 2009; Mattila and Tokola, 2019; Picchio et al., 2020; White et al., 2012; Murphy et al., 2007; Mohtashami et al., 2017; Uusitalo et al., 2020). Topographic modelling requires minimal input and the temporal variables used in the final model presented here, are freely available (Copernicus Climate Change Service, 2019). A spatiotemporal model predicting SWC could improve the guidance of machine operators in forest sites during harvesting operations, for example by the effective positioning of brush mats (Labelle et al., 2019; Labelle and Jaeger, 2018). Practical use of static, topographic maps has already been observed in Canada and Scandinavian countries (Ring et al., 2022). By incorporating a temporal aspect, the accuracy of these tools could be further improved. This has the potential to enhance sustainable forest management by protecting soil and mitigating harmful sediment transport (Ågren et al., 2015; Lidberg et al., 2020; Kuglerová et al., 2017; White et al., 2012).

### 4.2    Comparison to previous work on predictions of SWC

Since soil moisture predictions are crucial for a variety of forestry aspects, several publications have focused on this topic before. For example, Lidberg et al. (2020) predicted soil moisture classes using spatial models built on topographic indices, correctly classifying 73% of wet areas in a Swedish case study. Ågren et al. (2014) reported accurate predictions for 87-92% of observations by comparing soil moisture classes to DTW maps. Larson et al. (2022) used data from the Krycklan catchment and found an accuracy of 84% when comparing moisture classes to the recently developed 'SLU soil moisture map' (Ågren et al., 2021). However, these validations were based on static topographic maps. One attempt to make such static maps dynamic was realized within the DTW concept, which can be customized to calculate various scenarios to adjust to general moisture conditions (e.g., flow initiation areas of 0.25, 1, and 4 ha for wet, moist, and dry conditions, respectively), but selecting the most appropriate scenario during practical use can be a challenging task that requires significant expertise (White et al., 2012; Lidberg et al., 2020; Leach et al., 2017: 5432). To overcome this challenge, we aimed for improvement of soil moisture prediction and refined the spatiotemporal approach conceived by Schönauer et al. (2022). During cross-validation of IMT data from sites in Finland, Poland, and parts of the data used in this work, they reported an $R^2$ of 0.80. The models for the present study showed an $R^2$ of 0.759±0.136 (SSN) or 0.636±0.040 (IMT), corresponding to Kendall's τ of 0.710±0.095 or 0.620±0.016, respectively. Although this may not seem like an improvement, it should be noted that the data from German sites had less explanatory power of topography for predicting SWC. For example, DTW4 alone explained SWC to a very limited extent ($R^2$ = 0.037***).

### 4.3    Prediction of rutting

Besides the comparisons of SWC with DTW maps, various studies have also investigated the capability of topographic indices in predicting rutting – with conflicting outcomes. For example, Vega-Nieva et al. (2009) found that 65% of ruts deeper than 25 cm were located in areas with a DTW value of less than 1 m, and 93% of these ruts occurred in areas with DTW values less than 10 m. Similarly, Heppelmann et al. (2022) observed a high frequency of severe rut depth in areas with DTW values less than 1 m in Norway. However, Mohtashami et al. (2017) did not find evidence of such patterns in a field trial where the inclusion of DTW values did not improve the accuracy of a linear model to describe the extents and degrees of rut depth on machine operating trails. In agreement, Schönauer et al. (2021a) found no evidence that DTW or TWI could predict rut depth in a field trial conducted in a temperate broadleaved stand. In this study, we found a significant correlation between RD and $DTW025$ with a Kendall's correlation coefficient ($\tau$) of -0.52***. Yet, this correlation has to be seen with caution: It is mainly driven by differing ranges of RD between the two Trials, as can be seen in Figure C1A. We observed that the temporal adjustments of the model based on current moisture conditions improved predictions of rutting by up-to-date SWC predictions, leading to a $\tau$ of 0.61*** (Figure 6B,C). While a strong association between RD and predicted values of SWC was observed, the influence of differences between the trials is evident. However, the ranges of RD for each trial were consistent with the SWC predictions. In $Trial_{WET}$, a significant correlation between RD and $SWC_{PRED}$ was observed (Figure 6B). We hypothesize that the wetter conditions during this trial, which lead to soil destabilization (Hillel, 1998; McNabb et al., 2001), enhanced the predictive power of topographic indices representing soil water distributions. For instance, $DTW025$ overlapped with surface water in depressions, as observed in the field campaigns for $Trial_{WET}$.

In contrast, during $Trial_{DRY}$, no correlation was found between RD and $SWC_{PRED}$. SWC along the measuring sections was likely below the threshold for soils to become susceptible to deformation. For example, Poltorak et al. (2018) stated that ruts only occurred on soils with an SWC above 50%, whereas $SWC_{CORE}$ at $Trial_{DRY}$ was below 30% (Figure 5).

### 4.4    Description of the model

The best-performing model in predicting RD incorporated temporal information from $SWC_{ERA}L2$, Month and Season, as well as spatial information from $DTW025$, TWI, SPI and DTW4, and was based on data from the manual measurements (IMT). The IMT data was collected in close proximity to the rut depth measurements at Site A (Figure 2), or with a distance of up to 1.3 km at Site B. However, the spatial distance between the IMT training data and the rut depth data did not seem to be crucial for the accuracy of predicting rut depth (Figure B1), since Kendall's $\tau$ between RD and $SWC_{PRED}$ was similar for both sites. Surprisingly, the correlation between in-situ $SWC_{CORE}$, sampled directly at the machine operating trails, showed a lower explanatory power in predicting RD than $SWC_{PRED}$. Although an overall association was confirmed, no correlation could be found when trials were analysed individually.

#### 4.4.1    Temporal variaton was more important than spatial variation

The lacking association between RD and $SWC_{CORE}$ on individual trials indicates that the temporal variability in soil moisture between the trials was more important in this study than the spatial variability within the relatively small areas where each trial was conducted. The spatial distrubution of the rut depth measurements might have been limiting in the present work. The semivariogram indicates the spatial covariation of rut depth and SWC (Figure 7). While the covariation of RD in Site A is indicated to be high within a range of 10 m (RD-transects were at this distance), on Site B during wet conditions, the sill of

the semivariogram reaches almost 40 m, which covered a high number of transects. Similarly, excluding soil information in
the initial stages of feature reduction suggests homogeneous soil properties on the relatively small study area.
Therefore, we have to admit, that the study design was not ideal for assessing the ability to predict rutting with a spatiotemporal
model of SWC, and the results have to be considered with caution.

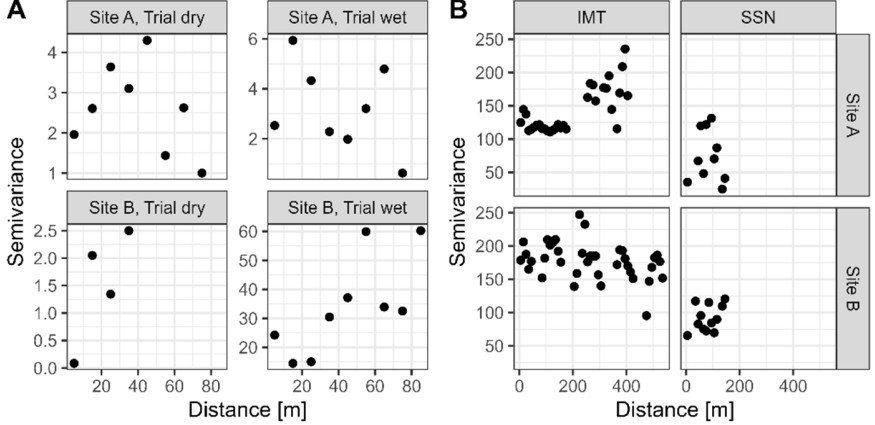


Figure 7. Semivariogram illustrating spatial autocorrelation of (A) rut depth (cm) and (B) soil water content (SWC) across the study area. Rut depth was measured during two moisture conditions, at four machine operating trail sections, allocated on two sites. The measuring transects had a spacing of 10 m. SWC was measured with handeld measuring techniques (IMT), or a soil sensor network (SSN) (Figure 2).

The spatiotemporal model (IMT), also supports the conclusion that spatial variations were either underrepresented by the
study design or very low compared to temporal variation by nature as the temporal feature $SWC_{ERA}L2$ was selected as most
important variable and the difference between the model with one predictor variable vs. the final model was small (Figure 4).
Still, this slight increase in the models' quality allowed for the integration of spatial patterns and resulted in the significant
but vague prediction of RD in Trial$_{WET}$ ($\tau = 0.344^*$, Figure 6). Another indication of the integration of spatial patterns can be
interpreted by the segregation of the temporal range of the IMT data (2019-2020) and the actual Trials (March 2021 and
October 2022, Figure 3), indicating a generalization of spatial and temporal patterns.
**4.4.2    Most important variables**
In the final model (IMT), $SWC_{ERA}L2$ has been identified as the most important variable, followed by Month and Season. It is
noteworthy that in the data with broader spatial coverage (i.e. IMT), in contrast to the SSN data, dynamic variables took
precedence over predictor variables. Surprisingly, when modelling SSN data, characterized by high temporal resolution and
low spatial resolution, DTW025 remained the most influential variable. One might have anticipated the opposite, expecting a
topographic index to play a central role in modelling IMT data, and dynamic $SWC_{ERA}$ variables dominating the modelling of
SSN data.
We presume that the low spatial variations of SWC in comparison to temporal variations, inadequately represented by the
provided topographic information, may have contributed to this unexpected outcome. Furthermore, the wider spatial coverage
in the IMT data likely resulted in more robust averages of SWC, leading to a stronger correlation with the coarse spatial data
of ERA5-Land (9x9 km). On the contrary, the SSN data, originating from areas with a size of 100x100 m and known for their
temporal wetness, could explain the heightened importance of DTW025. Some sensors might have measured constant water
saturation, thereby inflating the explanatory power of topographic information. These assumptions are speculative, and further
research in this direction is warranted.
In the feature reductions of IMT and SSN data (Figure 4), $SWC_{ERA}L2$ (7-28 cm soil depth) dominated over $SWC_{ERA}L1$ (0-7
cm). This aligns with in-situ measurements of SWC by the SSN, conducted at a soil depth of approximately 10 cm (Figure
3A). Even for the IMT data, where SWC was measured in the top 6 cm of soil, $SWC_{ERA}L2$ yielded a better goodness-of-fit
compared to $SWC_{ERA}L1$ (Figure 3B). We hypothesize that the prevalence of open lands as the dominant land cover form in
the ERA5-Land raster cell (section 2.2.4) contributed to the superior fit of $SWC_{ERA}L2$. Grasslands typically exhibit higher
temporal heterogeneity of soil moisture compared to forests (James et al., 2003). This temporal heterogeneity tends to decrease
with deeper soil layers (Tromp-van Meerveld and McDonnell, 2006). Therefore, the stronger correlation between $SWC_{ERA}L2$
and SWC, as well as its higher importance within the random forests, seems reasonable. The disparity between $SWC_{ERA}$ and
in-situ SWC can be attributed to the high transpiration rates in forests, as opposed to grass (Kelliher et al., 1993).

## 4.5    Further developments

The terrain data was derived from a digital elevation model, which is increasingly available for the entire Europe (Hoffmann
et al., 2022), while the dynamic variables are based on date and retrievals from ERA5-Land, which are freely available up to
a few days ago. These inputs would allow for automated mapping of current soil water content, which could be made
accessible to forestry stakeholders. Recent developments also show a pathway to integrate medium and long range weather
forestcasts into trafficability predictions, as conceived by the Finnish Meteorological Institute (2023). Both, recent as well as
forecasting predictions can lead to improved soil protection, higher efficiency of timber harvesting (Suvinen and Saarilahti,
2006), and a new stage of sustainable forest management (Campbell et al., 2013; D'Acqui et al., 2020; Uusitalo et al., 2019;
Jones and Arp, 2019). However, it should be noted that the in-situ data of SWC originated from manual measurements, and
it was relatively labor-intensive to gather this amount of data. There is potential to reach appropriate accuracy even with a
reduced dataset - further investigation would be necessary to determine the essential input data criteria. The alternative to
manual measurements is given by sensor networks, which led to comparable results, but such sensor networks are expensive
to establish and maintain. Nonetheless, initiatives of installing sensors are emerging and additional manual measurements
could be conducted. In the future, forestry stakeholders who require accurate raster predictions could potentially facilitate
manual measurements or install sensors and provide the captured data to scientific organizations, which could deliver
spatiotemporal soil moisture predictions in return. The captured data could be made available for creating spatiotemporal
models of SWC, allowing for additional training data and daily raster predictions for new areas of interest, with various
scientific insights and practical applications.

## Conclusion

In this study, we developed a spatiotemporal model that used multiple topographic indices, temporal variables, soil moisture
retrievals from ERA5-Land, and data from manual measurements to predict soil water content (SWC). Predicted values of
SWC were compared to rut depth data collected during four forwarder trials. Overall, the model performed well in predicting
rut depth, with a Kendall's $\tau$ of 0.61 for all trials. Yet, this result has to be considered with caution, since spatial covarition
was detected in parts. We hope, that this experience helps for future research, in which more attention to spatial covariaton
on soils should be paid. Still, we believe that a dynamic prediction of SWC will help forest managers and machine operators
avoid wet areas, leading to more sustainable forest operations. Using freely available temporal information is a significant
improvement, as it enables more accurate and up-to-date predictions, which allow to make more informed decisions and avoid
potential hazards. Future work should focus on developing automated pathways for generating daily raster predictions of
SWC, and on generating reliable and comprehensive in-situ data. There is a need for more data on rutting and SWC, measured
with a sufficient spatial coverage, whether by manual measurements, the establishment of additional sensor networks, or by
automatic ways of capturing rut depth data through machines driving off-road, to cover more areas and different sites and
regions.
**Data availability**
The data used in this work will be made accessible via Zenodo
**Author contribution**
MS and DJ designed the experiments and MS and FH carried them out. MS developed the model code and performed the
simulations. MS prepared the manuscript with contributions from all co-authors.
**Competing interests**
The authors declare that they have no conflict of interest.
**Acknowledgements**
We acknowledge the financial support from the Eva Mayr-Stihl Stiftung for this work. We extend our gratitude to the
Geological Survey of Northrhine-Westphalia (Landesbetrieb NRW) for conducting the soil mapping on the experimental sites
and for their contributions to the field trials analysis. In particular, we would like to thank Dr. Heinz Peter Schrey, Dirk Elhaus,
Thilo Simon, and Rainer Janssen. Our appreciation also goes to the Forest Education Centre, Forstliches Bildungszentrum,
Zentrum für Wald und Holzwirtschaft, Landesbetrieb Wald und Holz NRW, Arnsberg, Germany, for their valuable support
during the fieldwork. Special thanks to Thilo Wagner and Thomas Späthe for their efforts in organizing the field trials, and to
Michael Schulte for operating the forwarder. ChatGPT (OpenAI, San Francisco, CA, USA) provided assistance in sentence
editing – all content was generated solely by the authors.
**Funding**
This work was supported by the cooperation project "BefahrGut" funded by the State of North Rhine-Westphalia, Germany,
through its Forest Education Centre, Forstliches Bildungszentrum, Zentrum für Wald und Holzwirtschaft, Landesbetrieb Wald
und Holz NRW, Arnsberg, Germany; by the Bio Based Industries Joint Undertaking under the European Union's Horizon
2020 research and innovation program, TECH4EFFECT Knowledge and Technologies for Effective Wood Procurement—
project, [grant number 720757].

**Appendix A**

To model the dataset consisting of both IMT and SSN data, the procedure described in section 2 was followed. The IMT dataset was merged with a subsample of the SSN dataset, where the sample size of the SSN part was twice that of the IMT dataset. This was done to prevent over-weighting of the SSN dataset. The resulting combination of IMT and SSN data was called the "Mix" dataset.

The final model using the Mix dataset included the input variables $SWC_{ERA}L2$, Month, TWI, $SWC_{ERA}L1$, DTW025, Season, DTW1 and DTW4, and achieved a $\tau$ of 0.655±0.081 (which corresponded to $R^2$ values of 0.639±0.108). Figure A1 shows that the correlation between the model outputs ($SWC_{PRED}$) and rut depth (RD) was significant.

Since the models trained on the Mix dataset did not perform better than those trained on the IMT or SSN datasets, we did not investigate the fused data partition any further, as one research question addressed the use of different data origins. For future work, however, the fused data would provide additional information, as compared to the individual datasets.

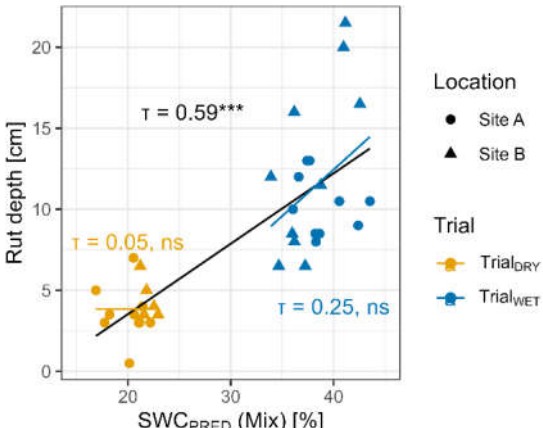

Figure A1: Rut depth (RD) was determined after four passes of a forwarder, driving on two Sites (A and B), during two seasons (Trial_WET and Trial_DRY). RD was compared to SWC values predicted by a random forest model trained on data from manual measurements or captured through a continuously measuring soil sensor network ('Mix'). Correlations were evaluated using Kendall's $\tau$ and significance levels are indicated by ∗∗∗ for p<0.001, ∗∗ for 0.001-0.01, ∗ for 0.01-0.05, (∗) for 0.05-0.10, and 'ns' for p>0.10.

   **Appendix B**

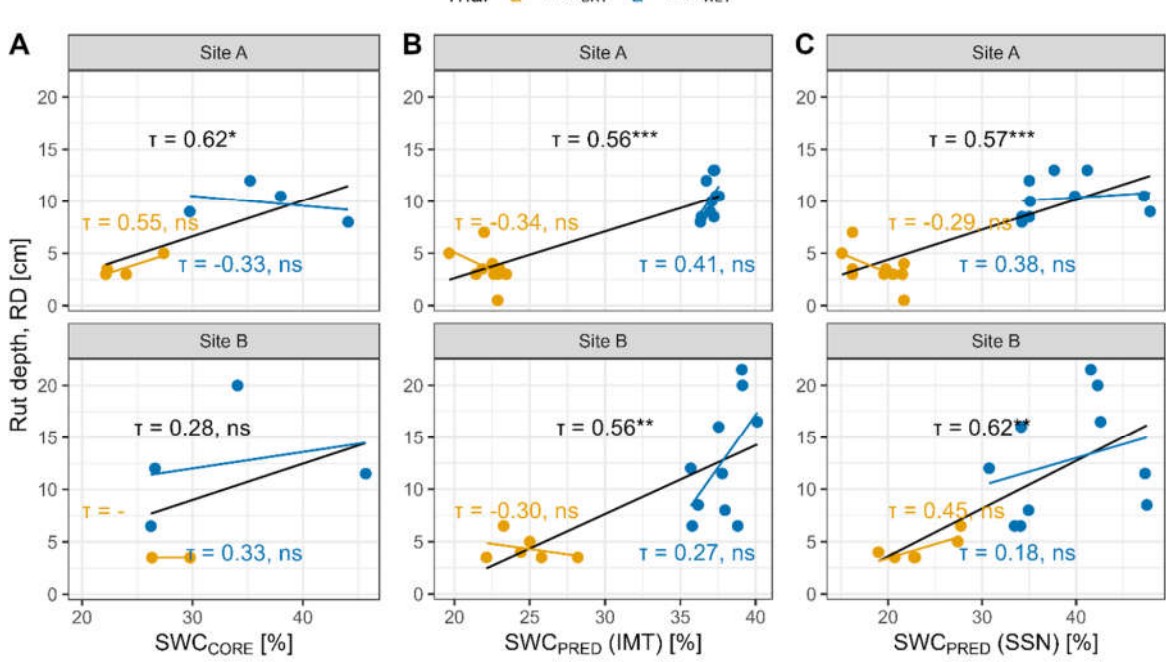

Figure B1. Rut depth (RD) was determined after four passes of a forwarder, driving on two Sites (A and B, Figure 2), during two seasons (Trial$_{WET}$ and Trial$_{DRY}$, conducted under different moisture conditions). RD was compared to SWC values, determined for undisturbed soil cores (A) and SWC values predicted by a random forest model trained on manually obtained IMT measurements (B, see Figure 1) and predicted by a model trained data from a continuously measuring soil sensor network (SSN, C). Correlations were evaluated using Kendall's τ. The correlation of all values is given in black, blue and yellow show the Trials during wet and dry conditions.Significance levels are indicated by * * * for p<0.001, * * for 0.001-0.01, * for 0.01-0.05, (*) for 0.05-0.10, and 'ns' for p>0.10.

475

**Appendix C**

Considering the significance of the topographic indices DTW and TWI in the development of the SWC models (Figure 4), we aimed to compare RD with both indices. Notably, RD exhibited a clear correlation with DTW025, the most conservative DTW scenario (Figure C1). TWI also demonstrated a correlation with RD.

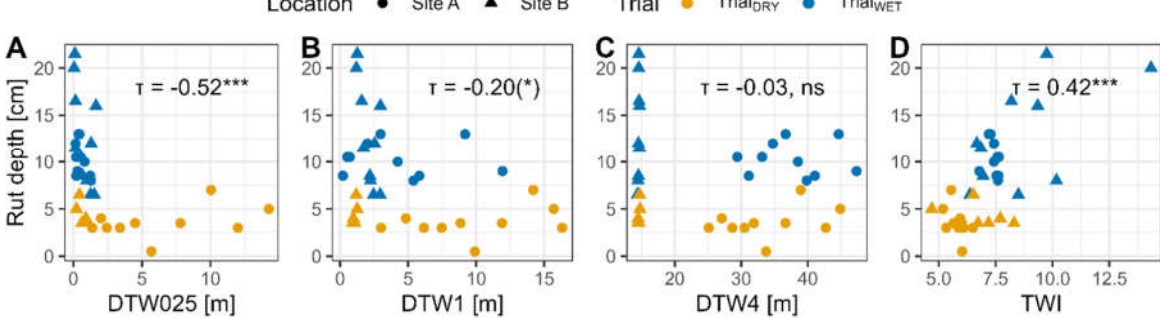

Figure C1: Rut depth (RD) was determined after four passes of a forwarder, driving on two Sites (A and B), during two conditions (Trial_WET and Trial_DRY). RD was compared to the topographic indices depth-to-water (DTW), calculated with different flow initiation areas (0.25 – 4.00 ha), and the topographic wetness index. Correlations were evaluated using Kendall's $\tau$ and significance levels are indicated by * * * for p<0.001, * * for 0.001-0.01, * for 0.01-0.05, (*) for 0.05-0.10, and 'ns' for p>0.10.

While showing significant correlations, the nature of these static maps does not allow for the representation of current moisture conditions. This limitation was overcome when using the predicted (or observed) values of SWC.

484

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
