# Peer review of "Soil moisture modelling with ERA5-Land retrievals, topographic indices and in-situ measurements and its use for predicting ruts"

_EGUsphere, 2023_

## Author Response (AR1)

**Author's Response**

Dear Editor Yongping Wei, dear Referees,

We extend our appreciation for your diligent review of our manuscript and the valuable feedback you provided. Your recommendations have significantly contributed to improving the overall quality of our work, and we have incorporated them into the revised version of the manuscript.

Best regards,
Marian Schönauer
On behalf of the authors

Please find our responses below, starting with a '#', and with line numbers [L…], referring to the 'clean version'. In response to your queries and comments, we have provided detailed responses earlier. We have built upon the information already conveyed in AC2 (https://doi.org/10.5194/egusphere-2023-1908-AC2) and AC3 (https://doi.org/10.5194/egusphere-2023-1908-AC3). For clarity and addressing any rebuttals, relevant portions of our previous responses are reiterated here for the corresponding issues.

> As indicated by the title and throughout the manuscript, this work is aimed at predicting rut depth with relevant predictors. However, no rut depth values were predicted. Instead, the model predicted soil moisture. The performance was then evaluated by investigating the correlation between the rut depth and the predicted moisture using Kendall's correlation coefficient.
> I do not think this is the correct approach. Either you need to predict rut depth directly and compare the predictions with the measured rut depth, or you build a second model to predict rut depth using the predicted soil moisture from the previous model and then compare the predicted rut depth with the observations. Simply showing the correlation between the predicted soil moisture and the measured rut depth is not enough, as this is not an apple-to-apple comparison. Given the above reasoning, the current work is incomplete.

**We recognize that the emphasis on predicting rut depth may have been too high. Therefore, we propose changing the title to 'Soil moisture modelling with ERA5-Land retrievals, topographic indices and in-situ measurements and its use for predicting ruts', and will present and discuss the results with more caution [e.g. L371ff, L426ff]. We seek the editor's and referee's understanding to retain this correlation, which is fundamental to our work.**

> ERA5 is a very coarse product with spatial resolution at kilometer level. Given the size of the study areas shown in Figure 7, I don't think ERA5 can provide enough useful information in terms of spatial variability. You may want to justify your selection.

**Thanks for this recommendation. We expanded the information about ERA5-Land and its advantages in the Material and Methods section [L161-165, L168-171]. A justification of choosing ERA5-Land and pivotal results can be found in the previous author's comments.**

The structure of some parts of the manuscript is very confusing. Most contents in the discussion section should be in the introduction section. You discuss results you got in the discussion section, not the motivation, existing efforts, and current gaps.

**We appreciate this comment and reworked the discussion in the revised version of the manuscript.**

Line 217: Why scale the τ values with 0.99 but not 0.98 or other numbers?

**We added a reference in L196 and discussed this threshold in the previos AC.**

The bottom part of Figure 2: As I said previously, the comparison between RD and SWC is not fair. Instead of a comparison, what was really investigated in this manuscript is the correlation between RD and predicted SWC.

**We have made adjustments to the figure, but would like to continue to compare and evaluate the correlations between rut depth (RD) and the predicted values of soil water content (SWC). The rationale for correlating RD with SWC has been discussed above, and we hope to have convincingly addressed any concerns raised by the referee [Figure 1, L101].**

I suppose some of your input data items may have different spatial resolutions. Maybe consider adding some explanation of how you unify them.

**Thanks for mentioning this. In principle, we just stacked different maps (with different resolutions), and extracted the raster values at spatial points (i.e. measuring or sensor positions) [L131ff, L172f].**

Similar to the one mentioned in the third major issue mentioned above, some titles in section 3 may not be appropriate and are not informative. For example, 3.3 Rut depth data. If it is more about the data, it shouldn't be in the result section.

**We fully agree and updated the headings. The rearrangement of parts of the M&M should provide additional clarity.**

Line 221: The usage of "Therefore" is confusing, as I did not see any causation between the previous sentence and the one that follows.

**Thanks for noticing. We have changed the phrasing of the sentences of regard [L234].**

Line 346: Wrong usage of "Despite". You talked about the advantages of the manually obtained dataset at the beginning and you recommended manual datasets at the end. I suppose "Therefore" is the correct word to use here.

**We changed the text as recommended [L400ff].**

For instance, what is the physical meaning of having DTW as an important predicting variable? Is this expected?

**We initially anticipated DTW to be one of the most influential predictors, but our assumption was partially disproven, as illustrated in Figure 4. We added new sections in the discussion, particularly in section 4.4 - subsections. For example, we discussed the physical meaning of DTW for both partitions, $SWC_{ERA}$ dominating the IMT modelling [L388-394].**

Why is DTW a better predictor in one case (IMT), but ERA-5 is the best predictor in the other case (SSN)? What are the differences in the observations that could lead to that?

**Please mind the new section 4.4.2 [L382-404].**

Also, predicted soil moisture is related with rutting depth only visually in a map.

**To enhance the connection between the figures (formerly Figure 6 and Figure 7), we integrated the maps illustrating raster predictions with scatterplots depicting the correlations between rutting depth (RD) and soil water content [Figure 7, L301].**

The actual relation between soil moisture and rutting depth needs to be further discussed, otherwise the title of the paper is inconsistent with what it delivers.

**We fully agree and emphasized the interrelations between soil moisture and soil deformation to address this concern in [L304-306, L40-42, L347-349]**

In site A, both different measurements overlap in space, so they are comparable. In site B, the different measurements do not overlap in space, so it's concerning to what extent they can be compared. This needs to at least be explained/discussed further in the manuscript.

**Certainly, we appreciate this observation. The IMT data was collected in close proximity to the rut depth measurements (Site 1), or with a distance of up to 1.3 km (Site 2) [L208-211, Figure 2]. However, the spatial distance between the IMT training data and the rut depth data did not seem to be crucial for the accuracy of predicting rut depth, since Kendall's τ between RD and $SWC_{PRED}$ was similar for both sites [Appendix B].**

L124, L246:  What is the explanation behind adding Year as an explanatory variable? I understand using Month and Season, as a given month or season could be consistently wetter than a different month or season. But it's not clear to me how a different year could be important to predict soil moisture. This could be relevant in a climate change study, when you have more than 50 years of data for analysis, for instance. Here, you only have a few years of data, which are not representative of extremes, so I am not sure I understand what is the reasoning in incorporating it as an explanatory variable. Moreover, it would be interesting to comment on the discussion on the impact that variables Month or Season could have in different climates (e.g., would they be relevant in places where temperature and precipitation are constant throughout the year?)

**As we agree with the referee's concern, we have removed Year from the predictor variables [L127].**

Figure 1: it looks like in Site1, IMT, SSN measurements and trials overlap spatially. In Site 2, it looks like the IMT measurements were performed in a different location than SSN + trials. This needs to be clearly stated in the methods section.

**We fully agree, and included information about the distance in the Material and Methods [L208-211]. We also addressed this issue in the results and discussion section [L357-360].**

In the results/discussion section, this issue needs to be addressed as well. **I would like to see the Kendall's correlations (e.g., Figure 5 and 6) drawn separately for site 1 and Site 2.** Maybe for Site 1, correlations could be better than Site 2, because of the spatial variations in sampling locations. It could be that you decide to focus on the results mainly from Site 1, because the measurements are more consistent there.

**This information has been added in Appendix B, and [L293-296].**

In the methods section, it's not clear to me the timing between different measurements and trials. IMT were collected monthly between Sep 2019 and Oct 2020 (L113). The SSN time coverage is not clear. The trials were conducted on Mar 2021 and Oct 2020 (L189 and L190). But then, in the results section, in Figure 3, it looks like trials were conducted in Mar 2021 and maybe Oct 2022?

Please state in the Methods section clearly the time coverage between each measurement and trial.

**Thank you for the hint. We added this information [L122-124] and corrected the typo [L206].**

Isn't it relevant that with SSN you have measurements during/after the trials, whereas for IMT, you only have measurements before the trials? This should be mentioned and the impacts of this different coverage in time should be discussed.

**Since the year as a predictor variable has been excluded as suggested, the temporal discrepancy should not be too important anymore (please mind [L379-381]). We believe that the temporal discrepancy is obvious from Figure 3 [L249]. Please mind AC3.**

Section 3.1: discuss for instance whether IMT and SSN are recording the same variable or not. According to the methods section, IMT measures soil moisture at 6cm, whereas SSN at 10cm. Is this a big difference or not?

**Both techniques are measuring the same parameter, volumetric soil water content – at least in principle. Certainly, the technology is different (refer to sensors in Figure 1,[L101]), but in this work, our focus was not on comparing the measuring results of both devices. Another difference arises from the depth of the measuring range. While soil depth can have a strong effect on soil water content (SWC), the vertical difference here is small, and we assume that the spatial heterogeneity of SWC is much larger than the difference of a few centimeters in soil depth. In addition to these factors, it should be noted that the modeling of SWC was conducted separately for each dataset (IMT and SSN), except for the mixed analysis in Appendix A. Therefore, we believe that potential differences between the in-situ measurements were not deemed too significant for this study and could not be verified. Nevertheless, a general agreement between SSN and IMT data can be observed in Figure 3 [L249].**

L159: How accurate is the ERA5-Land soil moisture retrieval? The resolution is 9 km x 9 km! So your site is mainly only covered by only one pixel. I think this mismatch in spatial resolution should be mentioned in the discussion when analyzing the results.

**ERA5 captures the 'regional' (9 x 9 km) temporal variability quite well, but within that region, there is substantial spatial soil moisture variability on a smaller scale. To address this, the main procedure of this work involved combining digital terrain indices that capture local-scale variability and merging that information with ERA5 data. This downscaling approach was employed to showcase a pathway for utilizing ERA5 retrievals for high-resolution predictions, considering the local-scale soil moisture variability not adequately captured by the original ERA5 resolution. We added information on land cover [L168-171] and its potential influence [L399-404].**

L161: Moreover, you stated that the ERA-5 is based on ground-based observations as well. I assume that the data is more accurate for regions where there are ground-based observations, and less accurate for regions where there are not. Was ground-based soil moisture data near your study sites used to "calibrate"/inform the ERA-5 product?

**We were not able to ascertain whether observation data from nearby stations was directly used by ERA5-Land. However, in response to this comment, the text has been corrected since 'ERA5-Land does not assimilate observations directly. The observations influence the land surface evolution via the atmospheric forcing. Forcing air temperature, humidity, and pressure are corrected using a daily lapse rate derived from ERA5' (Muñoz-Sabater et al., 2021) [L161-165]. We assume that even if a measuring site of ERA5-Land was 'nearby,' the spatial heterogeneity of soil water content (SWC) and differences between habitats would make it very unlikely that the distance between our study sites and the nearest ground-truthing site would have a strong influence on the accuracy.**

Figure 3: are these time series from Site 1, Site 2, or both? Please make this clear.

**The data originates from both sites, encompassing a total of 2 × 9 measuring positions. We have updated the figure caption accordingly [L250].**

Figure 3: based on the Kendall's coefficients, it looks like correlations in green/layer2/deeper soil are higher than correlations in blue/layer1/top soil. I think it's important to mention that in the results section. And to discuss it in the discussion session.

**That is correct. SWC$_{ERA}$ from layer2 (7-28 cm) ended up in a better representation of in-situ SWC than Layer1 [L396-404]].**

3.2. Soil moisture models: for SSN, the most important variable was DTW25 and for IMT, the most important variable was SWC$_{ERA}$. Can you explain why these variables turned out to be the most important ones? Was this what you expected? Are there physical reasons for that? You could maybe discuss this more in the Discussion section.

**Please find the response in the new section 4.4, particularly section 4.4.2 'Most important variables'. We appreciate the comment.**

L280-281: I think it's important to mention in the text that none of the models were significant for Trial 2. In the end, you made the decision of which model to use based on Trial 1.

**The correlation between the outputs of the SWC models and RD data was not utilized to select the final models. We aimed to improve clarity by re-arranging the method section and providing more details.**

It's important to discuss more in the Discussion about these differences in Trial 1 and Trial 2.

**Yes, we agree, and added the following information to the discussion [L345-353].**

Figure 7: I am not sure it is fair to use a model to predict soil moisture in trial 1 and trial 2, if you just used the data from trial 1 to train the model. Please discuss this.

**We did not incorporate data from the RD measurements in the modeling of SWC. In order to enhance clarity, we reworked and re-organized the sections in the M&M.**

The analysis of Figure 7 in general is a bit too shallow. Please discuss more the results of the model. Are predicted soil moisture and rut depth consistent? What is the correlation between these two variables (One way to check the correlation between two variables that have different units is to use the Spearman rank correlation. It takes into account the ranking between the different variables, not their absolute values.)

**We hope that the new Figure 7 [L301] is more intuitive compared to the previous one.**

Are the results of the model what you expected? Is the spatialization really relevant here (i.e., are there heterogeneities that the spatial model was able to represent?). And I ask this particularly because I didn't understand Figure 1: for site B, the measurements of IMT and SSN do not overlap. How did you compare them? If you compared them regardless of the spatial variability, maybe spatial variability is actually not that relevant here.

**While it would have been feasible to conduct cross-validation of the SWC models separately for each site, our primary focus was not on site-specific effects but rather on the general impact of topography on soil moisture dynamics. We asserted that both sites were relatively comparable in terms of soil and stand properties, and our objective was to leverage the varied positions of SSN sensors or IMT measurements to derive topographically driven patterns.**

**It's important to note that the IMT data was not directly compared with the SSN data. These datasets from different sources were treated independently in the modeling process. Consequently, this form of spatial variability does not appear relevant to our analysis. We hope that discussing this issue provides more clarity about the manuscript [L358-360].**

Can you use the results of the model to actually predict rut depth? If not, please at least discuss this and why not. Otherwise, I think the title of the paper needs to be updated, because in the end rut depth was not predicted as stated in the title.

**We updated the title and present the results with more caution [e.g. L371ff, L426ff, L1f].**

L315-320: you discuss the relation between rutting and DTW reported by other studies. But, in your study, what were the outcomes in terms of rutting depth and relation with DTW? This is not clear.

**We added the new section 3.3.1 [L277-283].**

There seems to be important differences between Trial 1 and Trial 2, and they have different conditions (wet/dry). Would precipitation data be useful as a proxy for rut depth as well? If yes, comment about this in the discussion. If not, just ignore this comment.

**Please mind AC3.**

In section 4.1. Importance of predictive systems, you mention the importance of "predictive systems", and, by that, I understand that one could use the methodology described in the paper to predict rut depth in the future. And one of the highlights of the methodology here is that it incorporates the temporal variability, by using the ERA-5 product. However, ERA-5 data could not be used to predict rutting depth in the future, given that the ERA-5 data is for the present, it is not a forecast for the future. And this brings back the question as to whether precipitation could be an important proxy, because, for precipitation, there are forecasts available.

**The term 'Prediction' used in this work excludes forecasts into the mid-range future but aims to predict rut depths that can be anticipated in a forest operation conducted today or tomorrow. While we have gained some experiences with forecasts of soil moisture content, we have observed that uncertainties can be quite high. Consequently, we decided to focus on current predictions rather than extrapolations with potentially high biases.**

**However, it's worth noting that there are attempts to extend predictions into medium-range forecasting, as demonstrated by efforts such as those by the Finnish Meteerological Institute [L408-412].**

L15-16: "(…) to model the response variable, in-situ soil water content" -> "(…) to model the soil water content" – I see no need for the "response variable", as it is a very vague term.

**Thank you for the hint, we omitted the "responsible variable" as recommended [L15f].**

L20: "(…) as well as temporal components such as numeric variables derived from date and (…)" – "numeric variables" is too vague.

**After removing Year as a predictor, Month and Season could be found in the final models. We have updated the abstract accordingly [L20].**

L34: I find it a bit strange to put a reference in a middle of a statement. "Soil compaction as a consequence of harvesting operations (Eliasson, 2005; etc) is detrimental to a (…)". What is the statement that these citations refer to?

**We rephrased the sentence [L33-36].**

Section 2.1.3. Soil maps: what is the information on these soil maps? Is it soil texture? Please state it.

**We used the (categorial) soil type information and added the information to the Materials and Methods section [L157].**

L60: Agren et al. (2021), used -> without the comma

**Done [L59].**

L91: when you mention the predictor variables here, it would be nice to have an idea of which variables are these. For instance, add in parenthesis (e.g., topographic indexes, soil texture)

**We have changed the statement as recommended [L91].**

I think it would make the paper much clearer if you added a sub-section in the beginning of Material and Methods called "Study design" or "Study overview". In this section, you could present Figure 2, and you write in a little bit more detail what is written as the caption of Figure 2. I think having an overview of the study design before reading the technicalities of the methods section would be very helpful.

**That is a good idea, we implemented a short overview to introduce the method section [L94-101].**

L110-111 and L117-118: both are "known to be temporally wet or sensitive machine traffic" – it's repetitive. Add this sentence only one time referring to both sites.

**Yes, that's true. We have removed the repetitions, thanks for mentioning.**

L143: what SPI stands for?

**stream power index [L146]**

L148: how is "Basin" a variable? Is it the basin area?

**We removed Basin from the predictor variables. (Also mind AC3).**

L154: how were you "able to gather maps"? What is the source for these maps?

**We adjusted the text accordingly [L156-159].**

L149-L150: I couldn't find any justification for re-sampling to 15m x 15m based on Agren et al. (2014). Please explain where this number comes from.

**We added a clarification of 15x15 m in the M&M section [L152-154].**

Also, in general, please add clarification on how datasets with different spatial resolutions were merged.

**In principle, we just stacked different maps (with different resolutions), and extracted the raster values at spatial points (i.e. measuring or sensor positions) [L132f, L172f].**

I think the methods section is too long. Some steps are described as a "recipe" instead of a scientific paper. For instance: L130: " (…) inserted into the attributed of a shapefile", L169: "(…) merged with in-situ data" – these small technical steps of merging/formatting two datasets don't need to be detailed.

**To provide sufficient information in the methods section is a delicate balance – between overly lengthy descriptions and insufficient detail. We aim to ensure that interested scientists can replicate the methods. Readers less interested in detail can skim through the text. We hope for the editor's and referee's approval.**

I would avoid using full sentences to describe the names of variables. For instance, in L138-139, "(…) of the following sizes: 0.25 ha (DTW025), 1.00ha (DTW1), 4.00 ha (DTW4)". In L 153-155: "(…) scale of 1:5,000 from forest site surveys (Soil05)." In L156-157, "(…) scale of 1:50,000 are available for the entirety of North Rhine-Westphalia (Soil50)"

**We agree and changed the explanations as recommended [e.g. L141].**

L159-162: I would rephrase it as: "ERA-5 Land is a global (…) , including soil moisture [$m^3$ $m^{-3}$] at the top soil layer (0-7cm) and at a depth of 7-28 cm. The soil moisture at the top soil layer is retrieved by assimilating satellite and ground-based observations". The names of the variables Volumetric soil water layer 1 and 2 are not relevant. I don't think you use them further in the manuscript. If you do, then you can add their names in parenthesis, but if not, I see no reason why they should be mentioned.

**We agree and rephrased the sentence, but kept the information on the layers, since it is needed for the feature selection [L161-165].**

In Figure 2, in the second row (predictors), there are some gray lines in the back, which seem to be connecting "soil maps" and the ERA graph to "add data". However, I assume "topogr. indices" and "Month Season" are supposed to be included as well?

**Certainly. We updated the flow chart (Figure 1, [L101]).**

Figure 3: a legend on the side with the colors and the names of the variables would be helpful.

**Legend was added, as suggested (Figure 3, [L249]).**

Figure 3: no need to say "the figure displays"

**This sentence part was removed, as recommended [L250].**

Figure 3: the names 'layer 1' and 'layer 2' are not really relevant here. I think it would be more relevant to provide what these layers refer to: layer 1 (top soil) and layer 2 (7-28cm depth).

**Changed as recommended [L250f].**

L232-233: "Soil water content was measured (…) August 2020" – I consider this fits in Methods, not Results.

**The sentence was removed. The abbreviations IMT and SSN are described, as repetition.**

2.2 Rut depth data: make it clear here what is different between Trials 1 and 2. Trial 1 is in a wet condition and Trial 2 is in a dry environment, right? That's why it is expected that SWC1 > SWC2. This makes the interpretation of Figure 6 afterwards easier.

**We changed the names of each Trial to Trial$_{WET}$ (=Trial 1) and Trial$_{DRY}$ (=Trial 2) to increase readability [L205-206].**

Figure 6: what does the black lines and the black values refer to? Make it clear in the caption.

**The caption was updated [L301f].**

L346: but in site 2, the locations of IMT and SSN are different.

**Certainly, this issue is addressed in the revised manuscript, as described above.**

L272: "$SWC_{PRED}$ proved to be a better predictor of rut depth", particularly for Trial 1 (in wetter conditions).

**We have changed that as recommended [L287f].**

L317: ground water -> groundwater

**This has been removed.**

L316-318: 65% + 93% is more than 100%. I don't think I understand this sentence. Please reformulate it.

L316-318: proximity to groundwater and DTW are two different things, no?

**The text has been changed [L335-337].**

Discussion: section 4.1. Importance of predictive systems: I think I would move this in the very end of the discussion.

We agree that swapping section 4.1 is an alternative but would like to stick with the current storyline. We hope for the editor's and referee's understanding.

We sincerely hope that the revisions made enhance the overall quality of the manuscript, and we look forward to the feedback from all parties involved.

**References**

Muñoz-Sabater, J., Dutra, E., Agustí-Panareda, A., Albergel, C., Arduini, G., Balsamo, G., et al. (2021). ERA5-Land: a state-of-the-art global reanalysis dataset for land applications. *Earth Syst. Sci. Data* 13, 4349–4383. doi: 10.5194/essd-13-4349-2021

---

## Author Response (AR2)

**Authors response:**
**Suggestions for revision or reasons for rejection (Referee 1)**

(visible to the public if the article is accepted and published)

The authors' answers to most of my questions and the corresponding changes in the manuscript are acceptable. However, there are a few items that I felt more explanations or changes are needed.

Dear Referee 1,

we would like to extend our gratitude for the efforts you put into reviewing our manuscript. Your detailed feedback and suggestions have been instrumental in refining our work. We are pleased to inform you that we have successfully integrated all of the recommended changes, which are briefly summarized below:

1. The added content is not enough to justify the use of ERA5 in this study. As pointed out by both reviewers, the resolution of the ERA5 product is too low given the size of the study area. Therefore, the most convincing reasons to use ERA5 could be either one or both of these

 a)  the soil moisture variability within the study area is negligible and
 b)  there are no other data sources applicable.

The authors mentioned the first reason in the response to the other reviewer but not in the revised paper. I believe it's important to add that in the manuscript to clarify any potential concerns of future readers.

We agree, that providing more information why chosing ERA5-Land over other datasets improves the rationale of this work. Therefore, we have added a paragraph in section 2.2.4 (L165-174), addressing both arguments (a&b).

2. The paragraph under Section 3.3. I appreciate the modifications the authors made. However, I insist on my previous point that this section should not be there unless a stronger connection to the results is presented. Since the goal of this paper is building models instead of purely doing some field measurements, the discussion about those measurements should be in the Data section. Unless the content discusses the impact of those measurements on the modeling results. Right now, the section is just about those measurements. Another way to work this around is to break this paragraph down and put those sentences near the more relevant discussions about results in the sub-sections that follow.

Thanks for your feedback. We endeavored to streamline the manuscript by: 1. Relocating one sub-paragraph (previously labeled as 3.3.1) to the Appendix and transferring the description regarding RD to the Material and Methods section. Consequently, the new Section 3.3 now encompasses "Comparisons of RD with $SWC_{CORE}$ and $SWC_{PRED}$". As the revised title suggests, a primary focus of our research centers on predicting RD through spatiotemporal modeling, with the potential to support day-to-day forestry operations and foster sustainable forest management.

3. It's fine if the authors insist on keeping the evaluation involving both the RD and the SWC, but more effort will then be needed to differentiate those two evaluations, as one is an apple-to-apple comparison, whereas the other one is not.

I would say keeping the existing evaluation between the predicted SWC and RD is probably fine, as the goal for this is to demonstrate the usefulness of the predictions. However, I will definitely expect more quantitative comparisons between the predicted SWC and SWC measurements, as this one tells us how good your model is and it is the main objective of this work.

Therefore, I suggest considering adding a few more other metrics, such as RMSE and NSE, to the latter for a more comprehensive evaluation.

We calculated and added the mentioned evaluation metrics and show them in Figure 5.

Once again, we sincerely appreciate your time and expertise in evaluating our manuscript.

Kind regards,
Marian Schönauer, on behalf of the authors

(visible to the public if the article is accepted and published)

**Authors response:**
**Suggestions for revision or reasons for rejection (Referee 2)**

I have reviewed the manuscript by Schönauer et al. for the second time, and I am honored that so many of my comments were considered useful by the authors and further implemented in the manuscript. I consider the paper much better now. Compared to the previous version, the updated manuscript:
- Is easier to follow (e.g., changing from "Trials 1 and 2" to "Trials Wet and Dry")
- Addressed and explained many methodological concerns (e.g., locations of measurements in Sites A and B)
- improved the discussion of the results (e.g., why was DTW025 an important predicting variable?)
- is more honest regarding the study limitations (e.g., spatial covariation detection).

Dear Referee 2,

We would like to extend our gratitude for the efforts you put into reviewing our manuscript. Your detailed feedback and suggestions have been instrumental in refining our work. We are pleased to inform you that we have successfully integrated nearly all of the recommended changes, which are briefly summarized below:

However, I still have some minor (mainly technical) concerns:

The following changes were integrated as is.

L34: and has shown to be -> and has been shown to be
L100: during this field trials -> during these field trials
Caption Figure 2: dryer -> drier
L120: dezember -> december
L206: dryer -> drier
L206: "subsequent section 2 of the same machine trail (...) (Figure 2, Site A), or in close proximity of section 1 (Site B)" – isn't it the other way around (Site A and B) ? Based on Figure 2, it looks like in Site A it was in close proximity (i.e., roughly parallel), and in Site B, it continued on the same trail.

Thanks for noticing, we changed the names accordingly.

Figure 5: the legend indicates that the y-axis is SWC_CORE, but the figure indicates the y-axis as SWC_SR

An artefact from an older version of the manuscript, which now has been updated.

Figure 7: why is the asterisk in parenthesis sometimes [0.28(*)], and other times not [0.34*] ? Captions of several figures: "Significance levels are indicated by *** for p<0.001, ** for 0.001-0.01, * for 0.01-0.05, (*) for 0.05-0.10, and 'ns' for p>0.10" – asterisk in parenthesis for 0.05-0.10, but not for the other ranges.

The asteriks in parenthesis "(*)" is indicating a, lets say trend, with an p-value between 0.05 and 0.10. "*" indicates significance (threshold of 0.05), with p-value of 0.01-0.05.

L294: too fragile ?

Initially we wanted to say „seem to be fragile", but now removed the „to" so it matches the meaning again.

In Appendix B, in one of the plots, Kendall's coefficient is NANA?

There were not enough values to calculate tau, we replaced the NANA, given by the created function through a `-`.

L177-178: "the main outputs when both datasets were combined can be seen in Appendix A"
-> this is a result, and I think it would be more appropriate to mention this in the results section.

> We agree, the sentence of regard has been shifted to the results.

L305: particle-to-particle

> Changed as recommended.

"Section 4.4.1 Temporal variation was higher than spatial variation": can you really claim that? Have you quantified temporal and spatial variation, in order to be able to state that one is "higher" than the other? In the text, it is clearer that you mean that the temporal variation was more important than the spatial variation.

> Absolutely, the subheading has been changed accordingly.

L364: the sentence is started with "this indicates". What is "this" referring to in this context? Please consider re-phrasing it. For instance, the sentence could be: "The temporal variability in soil moisture between the trials was more important in this study than the spatial variability within the relatively small areas where each trial was conducted".

> `This` was related to the previous paragraph, but we agree, it is better to readdress the topic again (Line 366).

L367: Sita A -> Site A

> Changed as recommended.

L374: wether?

> Changed to ‚either‘

L371-372: Please re-phrase this sentence in a more formal way: "Therefore, we have to admit, that the study design was not ideal for assessing the ability to predict rutting with a spatiotemporal model of SWC, and the results have to be considered with caution." – remove the part about "we have to admit", maybe add something on the lines of "limitations" of this study.

> We would like to stick to this rather personal comment, as it was really a mistake by us (or due to resource-limitations).

> Once again, we sincerely appreciate your time and expertise in evaluating our manuscript.

> Kind regards,
> Marian Schönauer, on behalf of the authors